# Quantification of meteorological conditions for rockfall triggers in Germany

**Katrin M. Nissen**[1], **Stefan Rupp**[2], **Thomas M. Kreuzer**[3], **Björn Guse**[4], **Bodo Damm**[2], **and Uwe Ulbrich**[1]

[1]Institute for Meteorology, Freie Universität Berlin, Berlin, Germany
[2]Institute for Applied Physical Geography, University of Vechta, Vechta, Germany
[3]Institute for Geography and Geology, University of Würzburg, Würzburg, Germany
[4]Section Hydrology, GFZ German Research Centre for Geoscience, Potsdam, Germany

**Correspondence:** Katrin Nissen (katrin.nissen@met.fu-berlin.de)

**Abstract.** A rockfall dataset for Germany is analysed with the objective of identifying the meteorological and hydrological (pre-)conditions that change the probability for such events in central Europe. The factors investigated in the analysis are precipitation amount and intensity, freeze–thaw cycles, and subsurface moisture. As there is no suitable observational dataset for all relevant subsurface moisture types (e.g. water in rock pores and cleft water) available, simulated soil moisture and a proxy for pore water are tested as substitutes. The potential triggering factors were analysed both for the day of the event and for the days leading up to it.

A logistic regression model was built, which considers individual potential triggering factors and their interactions. It is found that the most important factor influencing rockfall probability in the research area is the precipitation amount at the day of the event, but the water content of the ground on that day and freeze–thaw cycles in the days prior to the event also influence the hazard probability. Comparing simulated soil moisture and the pore-water proxy as predictors for rockfall reveals that the proxy, calculated as accumulated precipitation minus potential evaporation, performs slightly better in the statistical model.

Using the statistical model, the effects of meteorological conditions on rockfall probability in German low mountain ranges can be quantified. The model suggests that precipitation is most efficient when the pore-water content of the ground is high. An increase in daily precipitation from its local 50th percentile to its 90th percentile approximately doubles the probability for a rockfall event under median pore-water conditions. When the pore-water proxy is at its 95th percentile, the same increase in precipitation leads to a 4-fold increase in rockfall probability. The occurrence of a freeze–thaw cycle in the preceding days increases the rockfall hazard by about 50 %. The most critical combination can therefore be expected in winter and at the beginning of spring after a freeze–thaw transition, which is followed by a day with high precipitation amounts and takes place in a region preconditioned by a high level of subsurface moisture.

## 1 Introduction

Landslides are geomorphological hazards associated with damage and fatalities to people and their connected structures (Froude and Petley, 2018). There is scientific consensus that specific weather conditions can strongly influence landslide occurrences (McColl, 2015). Thus, as the effects of climate change become more and more visible, the scientific community tries to understand and predict the consequences for landslides (e.g. Gariano and Guzzetti, 2016; Macciotta et al., 2017; Haque et al., 2019; Bajni et al., 2021). However, specific weather conditions must meet specific ground conditions for landslides to occur. Consequently, meteorological parameters and thresholds are spatially heterogeneous, and results from previous studies on this issue are site-specific (Spadari et al., 2012; Siddique et al., 2019). Furthermore, the term "landslides" encompasses multiple mass-wasting processes on slopes (e.g. mud flow and rockfall) that each depend on different preconditions and trigger mechanisms

(Varnes, 1978; Hungr et al., 2014). It is therefore sensible to study these different types of processes separately.

Against this background, the present study focuses on multiple rockfall clusters spanning all of Germany. Rock-
fall is the removal of superficial and individual rocks from a rock cut slope (Robbins et al., 2021). In solid rock, the density and size of fissures and cracks are preconditions that promote rockfall; i.e. they represent weak points vulnerable to weathering that may eventually dislodge individual rocks
(Erismann and Abele, 2001). Thus, all weathering mechanisms that promote rockfall can also be trigger mechanisms that cause the start of a rockfall event. Weathering mechanisms driven by meteorological events can be the wetting and drying of matrix pores in sand- and siltstones from precipita-
tion and evaporation (e.g. by means of swelling clays), carbonate dissolution in carbonatic rocks from rainfall, and frost shattering due to water-filled rock discontinuities at low temperatures (Souleymane et al., 2008; Krautblatter et al., 2012; Viles, 2013). In the case of frost shattering, weathering and
triggering mechanisms may differ, since rockfalls may occur during thawing rather than during cooling periods; this is because the cohesion of the ice–rock interface can be sufficient to hold the rock in place (D'Amato et al., 2016). The direct effect of temperature on the frequency of rock-slope
failures in permafrost locations is based on the same mechanism (e.g. Paranunzio et al., 2019; Savi et al., 2021), but does not play a role in our study region. Moreover, there are weathering mechanisms not directly linked to meteorological events that may promote or trigger rockfalls. Tectonic activ-
ity may weather rock through earthquakes, phases of folding, thrusting, strike-slip, and normal faulting (Di Luzio et al., 2020). Tree root growth may expand rock fractures and joints (Dorren et al., 2007). Lastly, anthropogenically induced vibrations and tremors (e.g. by explosions or machine use)
or direct constructional interventions may lead to weathering of rock (Gill and Malamud, 2017).

In this context, the question arises as to whether a statistical model focused on meteorological parameters can accurately predict rockfall occurrence. A first investigation con-
40 ducted on a monthly basis by Rupp and Damm (2020) already suggests that a relationship between rockfall events, temperature and precipitation is likely to exist in the selected study areas. The present analysis focuses on the quantification of these effects for a later application in climate change
studies. For this type of application, it is essential to consider all climatic factors that promote or suppress rockfall together, as they can reinforce or cancel each other out (Crozier, 2010). For example, climate projections suggest that heavy precipitation – a well-known rockfall promoter and trigger –
may increase in magnitude and frequency due to the higher moisture-holding capacity of warmer air (IPCC, 2014). At the same time, increases in evaporation due to higher temperatures decrease water availability, which may slow down weathering mechanisms in some cases. To account for this, a
statistical model that includes the interaction between the rel-

evant local daily variables was developed. The evaluation of simulated soil moisture and a pore-water proxy that accounts for evaporation as parameters for water availability distinguishes our approach from similar studies (e.g. Bajni et al., 2021; D'Amato et al., 2016; Macciotta et al., 2017; Sass and  60 Oberlechner, 2012).

## 2 Data

### 2.1 Rockfall

The present study uses historical rockfall data that are extracted from the landslide database of Germany (see Damm  65 and Klose, 2015; Rupp and Damm, 2020). Scientific publications, governmental reports, police reports, civil protection reports, newspapers, field data collections, and GIS and web analyses were the information sources for the landslide database, which currently contains about 6000 mass move-  70 ment events of different types. The database mainly covers the last 200 years, with the oldest event dated to 1137. Information on 670 rockfall events (Fig. 1) is included in the rockfall dataset. The focus of the present study is on a number of geomorphological processes (e.g. rockfall, rock topple, de-  75 bris fall, debris topple) that are characterised by the rapid gravitational downslope fall of debris or rocks (Whalley, 1974; Varnes, 1978; Flageollet and Weber, 1996; Sass and Oberlechner, 2012). Due to the different particle sizes and volumes of the detached masses, the mentioned processes  80 are subsumed under the generic term rockfall in this study (Evans and Hungr, 1993; Selby, 1993). In addition to an identification number, the location (i.e. coordinates) and the date of occurrence for each rockfall event are stored in the dataset. For a total of 343 (642) rockfalls the day (year) of occurrence  85 is known, while the remaining are undated. The time span of rockfall occurrences ranges between 1480 and 2018, with the majority of them ($n = 621$) recorded from 1873 onwards. It is important to note that the rockfall database is not comprehensive. The increase in the number of recorded events with  90 time (Fig. 2) is not due to climatic conditions but reflects the fact that data on rockfall events were more readily available in recent years.

The dense spatial clustering of rockfall events and high temporal data homogeneity guide the selection of three study  95 areas (Fig. 1; ES is the German part of the Elbe Sandstone mountains, HL is northern Hesse and southern Lower Saxony, and HR is western Hesse and Rhineland-Palatinate).

#### 2.1.1 Elbe Sandstone cluster

The ES cluster mainly includes the German parts of the Elbe  100 Sandstone Mountains, which are located on both sides of the upper reach of the river Elbe between the Czech city Děčín and the Saxon city Pirna. Geologically, the area is dominated by compact Cretaceous sandstones. Fracturing and formation of cracks and fissures came about by extensive uplift pro-  105

cesses and long-term tectonic stresses. Fluvial incision accounted for a heavily dissected relief with numerous horizontal cracks, vertical joints and clefts, and small gorges (Pälchen and Walter, 2008). The climatic conditions are characterised as continental, with warm summers and cold winters. The mean monthly temperature is between $-0.8\,°\mathrm{C}$ in January and $17.8\,°\mathrm{C}$ in July. Between 1946 and 2017, annual precipitation ranged between 398 and 1153 mm with an annual average of 758 mm.

### 2.1.2 Hesse and Lower Saxony cluster

The HL cluster embeds large parts of the northern German Central Uplands, i.e. the Hesse Highlands and Lower Saxon Hills. Predominantly, the geological conditions are characterised by Middle Lower Triassic Bunter Sandstone. Pronounced dissections were caused by tectonic stresses (Damm et al., 2010). Quaternary sediments, for example periglacial cover beds and loess covers, cover the bedrock in large parts of the area (Wagner, 2011; Damm et al., 2013). The climate can be described as temperate with warm summers and mild winters. The mean monthly temperature is between $0.5\,°\mathrm{C}$ in January and $17.3\,°\mathrm{C}$ in July. From 1902 to 2017, annual precipitation ranged between 357 and 1099 mm with an annual average of 660 mm.

### 2.1.3 Hesse and Rhineland-Palatinate cluster

The HR cluster comprises large parts of the Hunsrück Hills in Rhineland-Palatinate and a small part of the Taunus Hills in Hesse. Geologically, Devonian bedrock, namely slate and quartzite, is predominantly present in this area. Distinct plateaus alternate with ridges and incised valleys (LGB, 2005). The climate can be described as temperate, with mild winters and warm summers. The mean monthly temperature is between $1.4\,°\mathrm{C}$ in January and $18.4\,°\mathrm{C}$ in July. Between 1915 and 2017, annual precipitation ranged between 324 and 853 mm with an annual average of 641 mm.

## 2.2 Meteorological and hydrological variables

For this study, datasets with a long record and high horizontal resolution were used in order to identify meteorological and hydrological conditions for as many rockfall events as possible with sufficient accuracy. It was therefore decided to use the gridded REGNIE dataset (Rauthe et al., 2013) for daily precipitation amounts. The dataset is compiled from spatially interpolated gauge measurements of the quality-controlled German weather service (Deutscher Wetterdienst, DWD) stations. REGNIE is available since 1931 for western Germany. For the new (post-reunification) federal states, the time series starts in 1951. The grid boxes have a size of $1\,\mathrm{km}^2$.

In order to study precipitation intensities, the gridded radar-based climatology RADKLIM (Winterrath et al., 2018) was used. The dataset includes hourly precipitation from radar measurements adjusted to station observations and has a horizontal resolution of $1\,\mathrm{km} \times 1\,\mathrm{km}$. For the present study, the daily maxima were extracted. The time series is comparatively short, as it only starts in 2001.

For temperature, it was decided to use the gridded E-OBS dataset (Cornes et al., 2018) as it goes back to the year 1950. The horizontal resolution of the grid is $0.1° \times 0.1°$, which corresponds to approximately $7\,\mathrm{km} \times 11\,\mathrm{km}$ in Germany. For the analysis of freeze–thaw cycles, the ensemble mean of near-surface atmospheric daily minimum and daily maximum temperatures provided in the v21.0e version of the E-OBS dataset was used. A freeze–thaw cycle was defined as the transition from a daily minimum temperature below $-1\,°\mathrm{C}$ to a daily maximum temperature higher than $0\,°\mathrm{C}$.

The subsurface water content (e.g. soil moisture, cleft water, water in matrix pores) is measured generally only at very few sites. Spatially consistent soil moisture monitoring in Germany, for example, relies on modelled soil moisture (Zink et al., 2016). In this study we attempt to utilise modelled soil moisture as a representative for all types of subsurface water. We analyse the results of a simulation with the state-of-the-art, grid-based hydrological model mHM (Samaniego et al., 2010), which was calibrated at hydrological stations using daily time series of observed discharge. The model has a daily time step, a horizontal resolution of $5\,\mathrm{km} \times 5\,\mathrm{km}$ and six vertical levels from the surface to a depth of approximately 1.8 m. The hydrological model mHM considers different soil types. Each soil type has different soil layers and thus site-specific soil characteristics such as substrate distribution and hydraulic conductivity. The infiltration from the surface into the ground depends on these soil characteristics. The set-up is based on European datasets as described in Rakovec et al. (2016) and Samaniego et al. (2019). We analysed the relative moisture content (i.e. degree of saturation) for the entire column from the surface to a depth of approximately 1.8 m. It is common practice for this model to further normalise these values using percentiles (Zink et al., 2016) as the variability of the modelled values is too low.

With respect to our aim to develop a statistical model that can be used to analyse the rockfall probability under climate change conditions, a challenging point of using simulated soil moisture is that it is stored only for some climate scenario simulations. Additionally, the moisture variables and the depth levels they represent differ between climate models. Therefore, the usage of a pore-water proxy ($D$) as an alternative to simulated soil moisture was tested as a predictor for the logistic regression model. $D$ is defined as the difference between precipitation accumulated over a period of time ($\mathrm{Prec_{acc}}$) and the potential evapotranspiration (PET) during this period:

$$D = \mathrm{Prec_{acc}} - \mathrm{PET}. \tag{1}$$

The term $D$ is also the basis for the calculation of the standardised precipitation evapotranspiration index (SPEI; Vicente-Serrano et al., 2010), which includes a standardisation of $D$ in order to allow comparisons between different

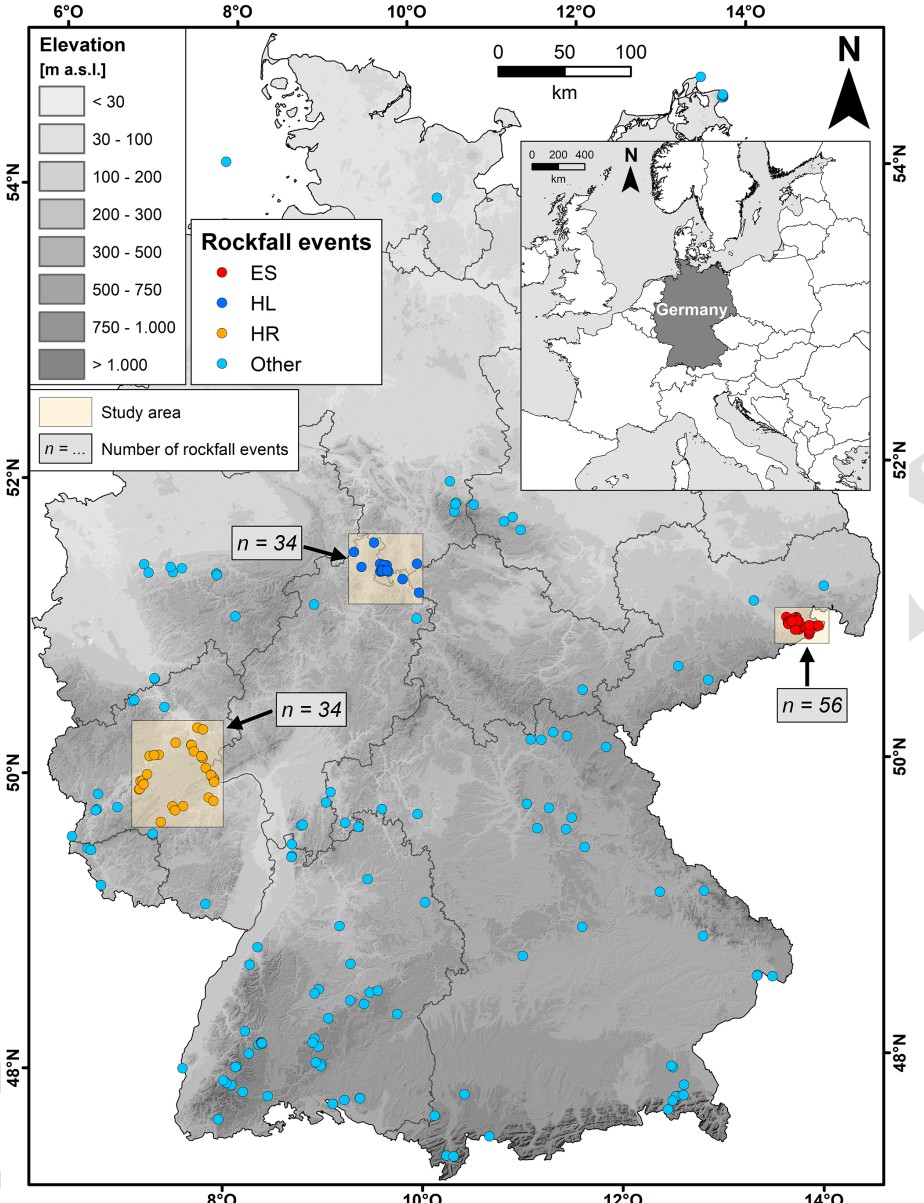

**Figure 1.** Location of rockfall events analysed in this study. Three distinct clusters – ES (Elbe Sandstone), HL (Hesse and Lower Saxony) and HR (Hesse and Rhineland-Palatinate) – are marked in red, blue and orange, respectively. All other events are coloured in light blue.

climatic regions, which is not necessary here. Different empirical methods exist to determine potential evaporation. In this study, the method first proposed by Hargreaves (1994) in the version modified by Droogers and Allen (2002) was applied. As input parameters it needs extraterrestrial radiation (which depends on latitude and day of the year), the period mean of maximum and minimum daily temperatures, and mean precipitation over the period of interest (which is used as a proxy for cloudiness). $D$ does not depend on the material of the ground and is an indicator of general water availability, thus accounting for water in rock discontinuities as well as in matrix pores. Therefore, further mentions of

pore water include water in discontinuities if not specified otherwise.

A relationship between the triggers and events can only be established for the sites and periods for which both elements are known. Thus, the analyses carried out in this paper include only data from grid boxes that contain the site (es) of at least one rockfall event occurring within the observational period of the respective record. Percentiles for soil moisture and $D$ are determined using the observations at these sites rather than all grid points within Germany. We refer to them as "across-site" percentiles.

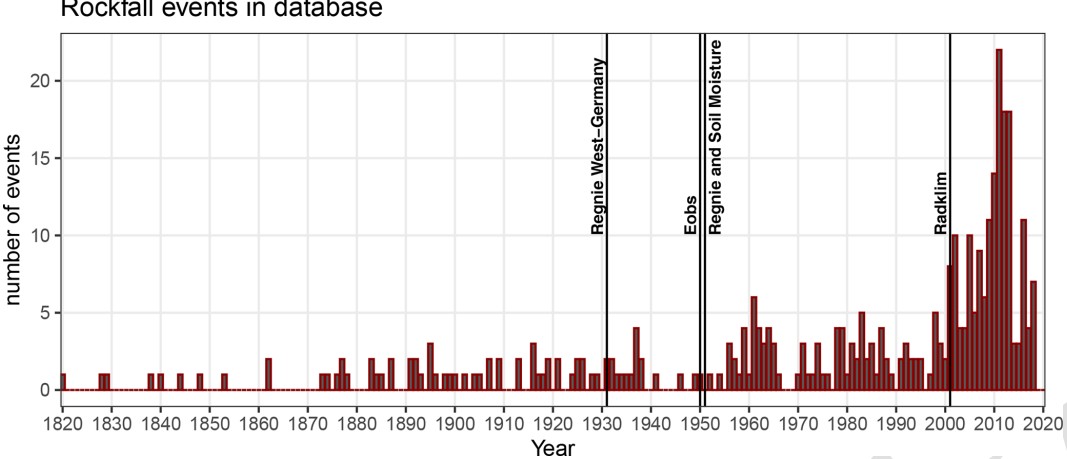

**Figure 2.** Time series of the number of rockfall events per year included in the database. The years at which the meteorological and hydrological observations start are indicated.

## 3   Methods

### 3.1   Weight of evidence

Weight of evidence (WOE) can be used to describe the relationship between an independent and a dependent variable and to rank the predictive power of different independent variables (e.g. Neuhäuser and Terhorst, 2007). It is defined as the logarithm of the Bayes factor (Good, 1985):

$$\text{WOE} = \ln \frac{f(X\bar{Y} = 1)}{f(X\bar{Y} = 0)}, \tag{2}$$

where $X$ is the independent variable and $Y$ is the binary information of whether the event occurred or not. $f(X\bar{Y})$ denotes the conditional probability density function for $X$ if $Y$ is true ($= 1$) or false ($= 0$).

In practice, a continuous independent variable (e.g. precipitation amount) is split into bins containing an equal number of observations. The WOE for each bin (b) is then calculated separately. It depends on the fraction of days with an event (here rockfall) to that of uneventful days. For categorical variables the WOE is determined for each category.

$$\text{WOE}_b = \ln \left( \frac{\%\text{NoRockfall}_b}{\%\text{Rockfall}_b} \right) \tag{3}$$

An integral measure for the strength of the relationship between the dependent and independent variable is the information value (IV; Siddiqi, 2006). It is calculated as

$$\text{IV} = \sum_{b=1}^{\text{nbins}} (\%\text{NoRockfall}_b - \%\text{Rockfall}_b) \times \text{WOE}_b, \tag{4}$$

with nbins being the number of bins. According to Siddiqi (2006), a predictor is not useful for statistical modelling if the IV value is less than 0.02. The IV is also used to rank the variables according to their influence.

### 3.2   Logistic regression

Logistic regression is used to model the relationship between predictor variables and the probability of a binary response variable. For logistic regression, a generalised linear model with a logit link function is fitted (Wilks, 2011). The probability $p$ of the binary event (e.g. rockfall yes/no) can be expressed as

$$p = \frac{1}{1 + \exp(-b_0 - b_1 x_1 - \ldots - b_i x_i)}, \tag{5}$$

where $x_1, \ldots, x_i$ are the predictors (i.e. meteorological and hydrological variables) and $b_0, b_1 \ldots, b_i$ are the regression coefficients. The regression coefficients are determined by maximising the log likelihood. For this study, the glm function of the statistical software R (R Core Team, 2018) was used for this task.

### 3.3   Model validation

The classical score to compare logistic regression models of different complexity is the Brier skill score, which, however, becomes unstable for rare events such as ours (Benedetti, 2010). We therefore use the logarithmic skill score (LSS) instead, which behaves similarly to the Brier skill score but performs better for extreme probabilities (Benedetti, 2010; Wilks, 2011). The logarithmic skill score quantifies the percentage gain of using the statistical model over just predicting the climatological probability and is calculated as follows:

$$\text{LSS} = 1 - \frac{\text{LS}}{\text{LS}_{\text{clim}}} \cdot 100\%, \tag{6}$$

TS1 where $\text{LS} = \frac{1}{n} \sum_{k=1}^{n} \text{LS}_k$ is the logarithmic score, with $n$ being the number of forecasts and $k$ indicating an individual forecast.

The value of $LS_k$ is determined using the forecasted probability $p_k$ calculated by the logistic regression model.

$$LS_k = \begin{cases} -\ln(p_k), & \text{if rockfall event occurs} \\ -\ln(1 - p_k), & \text{if rockfall event does not occur} \end{cases}$$

$LS_{clim}$ is calculated analogously using the climatological probabilities $\frac{\text{events}}{n}$ (number of events per number of forecasts).

When comparing two statistical models predicting the same $n$ situations, a higher logarithmic skill score indicates better predictions.

Another option for comparing statistical models that were fitted based on the same observations is the Akaike information criterion (AIC; Akaike, 1974), which estimates the prediction error:

$$AIC = 2i - 2\ln(L), \tag{7}$$

where $L$ is the likelihood and $i$ is the number of predictors. Here, a lower AIC value is associated with the better model. The risk of overfitting is considered by penalising a high number of predictors.

Ensuring that no overfitting takes place can also be achieved by cross validation, which tests the statistical model on a sample of independent data. For this study, the full event catalogue was divided into five approximately equally sized groups, with events from the different clusters equally distributed between the groups. The statistical model was then trained using only four of the groups and afterwards applied to predict event probabilities in the remaining group. The logarithmic skill score for that group was calculated. The process was repeated for all groups, and a mean cross-validated logarithmic skill score was determined ($LSS_{cv}$).

## 4 Results

### 4.1 Selection of potential predictors

The weight of evidence analysis is used to analyse the potential of different predictors to influence rockfall probability. All variables were screened individually. Figure 3 shows the most robust estimate possible for each variable. It is based on all rockfall events that occurred during the observation period of the respective dataset and all unique observational time series at the location of these events. A graphical inspection of the result already reveals that a relationship between the independent variable and the probability of rockfall exists for all variables. Moreover, the IV value is higher than 0.02 for all variables.

For a consistent comparison of the IV values, the analysis was repeated with the number of grid boxes, time steps and events reduced to the subset covered by all datasets (see the Supplement). This slightly increases the IV values for daily precipitation and soil moisture. The highest IV in the

short common period (2001–2013) is obtained for daily precipitation (IV = 0.85). Soil moisture and precipitation intensity have similar IV values of 0.25 and 0.23, respectively, followed by freeze–thaw cycles (IV = 0.05). To take into account that the thawing process might take several days, a time span preceding the event was evaluated. Comparing different time spans, it turned out that the IV value associated with a freeze–thaw cycle immediately before the rockfall event (i.e. the preceding 2 d) was too low to be considered useful for statistical modelling (see Fig. S2a in the Supplement). Extending the analysis period backward in time increased the IV value, with a peak reached after 9 d (Fig. 4). The WOE analysis also confirmed that, in accordance with the findings of D'Amato et al. (2016), thawing increases rockfall probability while freezing decreases it (see Fig. S2).

### 4.2 Construction of a statistical model

Logistic regression is a well-established statistical method to determine probabilities for a binary event (e.g. rockfall vs. no rockfall) based on the conditions of independent variables. Here, a logistic regression model using precipitation, soil moisture or the pore-water proxy $D$, and freeze–thaw cycles as independent meteorological and hydrological variables is fitted. The consideration of individual rockfall clusters (Sect. 2.1) provides information on possible regional differences of the results.

The logistic regression models were fitted using $n = \text{es} \cdot \text{ts}$ data points, with ts being the number of days for which meteorological and hydrological data are jointly available among all variables used as independent parameters in the model. The number of event sites (es) at which a rockfall event was recorded within the period covered by the meteorological and hydrological observations depends on ts. Other than for the WOE analysis, we neglected the fact that some grid boxes enclose the site of more than one event and did not merge these sites (thus, es is events). The background is that the logarithmic skill score and the Akaike information criterion can only be used to compare statistical model alternatives if they are based on the same data points $n$. Due to the different spatial resolutions of the individual meteorological datasets, merging would change $n$ for each new combination of input variables. $n$ is only reduced for evaluations involving precipitation intensity that are carried out based on a much shorter period and a lower number of sites than all other evaluations. This will be considered when comparing the results.

To find the best performing statistical model, numerous combinations of the potential predictors were compared. Table 1 lists the results for a selection of these tests. Evaluated predictors include daily precipitation (precip_1day), the local percentile of daily precipitation calculated using wet days (precip_1day_lperc), across-site percentile of simulated total column soil moisture (sm_perc), across-site percentile of parameterised pore water ($D$_perc), hourly precipitation (pre-

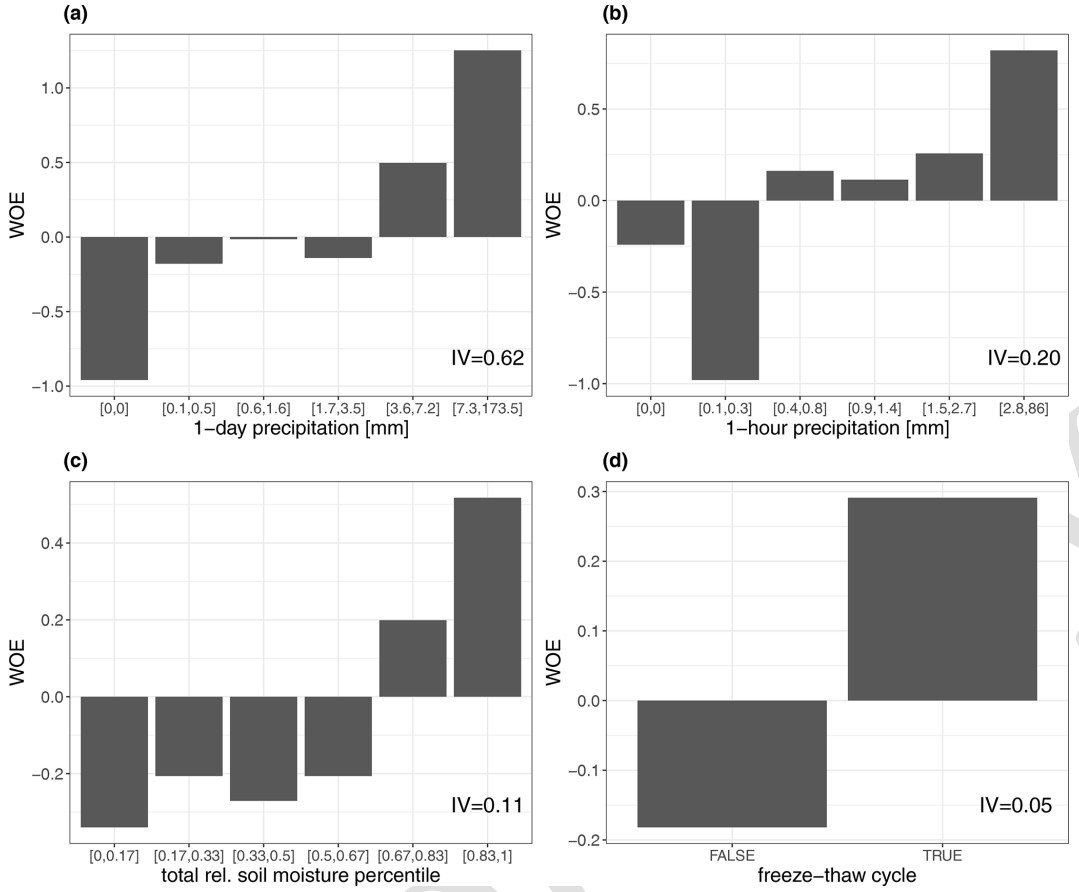

**Figure 3.** Weight of evidence (WOE) for **(a)** daily precipitation, **(b)** hourly precipitation, **(c)** percentile of relative simulated soil moisture content over all layers and CE1 **(d)** occurrence of a freeze–thaw cycle in the previous 9 d.

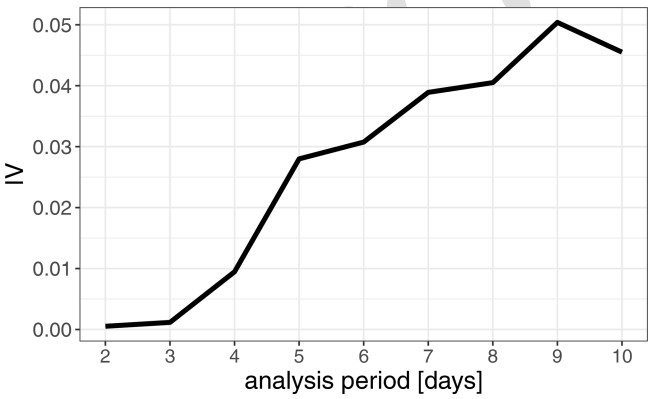

**Figure 4.** Dependence of the IV for freeze–thaw cycles on the period used for the analysis.

cip_1hr), the local percentile of hourly precipitation calculated using wet hours (precip_1hr_lperc), a categorical predictor denoting to which cluster an event belongs (cluster) and a binary predictor indicating if a freeze–thaw cycle occurred at the site during the previous 9 d (ftc). As the full

notation of the model equations is space consuming, we use the compact symbolic form used in the R programming language (R Core Team, 2018). The operator + denotes adding another predictor term, : marks the product between two predictors and ∗ TS3 indicates that all possible combinations of interactions between the predictors are considered. Thus, the term (precip_1day_lperc∗D_perc) + ftc in row 16 of Table 1 translates to TS4

$$p_k = \frac{1}{1 + \exp(\underbrace{-\beta_0 - \beta_1 \text{precip\_1day\_lperc}_k - \beta_2 D\_\text{perc}_k}_{\text{if ftc}_k = \text{TRUE}} - \underbrace{\beta_3 \text{ftc}_k}_{} - \beta_4 \text{precip\_1day\_lperc}_k D\_\text{perc}_k)} . \quad (8)$$

The performance of the statistical models listed in Table 1 is compared with the help of the cross-validated logarithmic skill score (LSS$_{cv}$) and the Akaike information criterion (AIC). A scientifically sound comparison is possible between models with identical es. Comparing models 1–3 shows that daily precipitation is more important than soil moisture and performs best if included in the form of its local percentile (denoted by the suffix "_lperc"). For hourly precipitation, LSS$_{cv}$ and AIC indicate that absolute values

**Table 1.** Symbolic formulae for a list of logistic regression models tested in this study and main characteristics associated with these models. The characteristics include the number of coefficients that needed to be determined, the number of event sites (es) that were used for fitting, the logarithmic skill score (LSS), the logarithmic skill score determined by cross validation ($LSS_{cv}$) and the Akaike information criterion (AIC). See text for explanation of symbolic equation notation.

|    | Symbolic equation | Coefficients | es | LSS | $LSS_{cv}$ | AIC |
|----|-------------------|--------------|-----|------|-----------|--------|
| 1  | precip_1day | 2 | 237 | 2.20 | 2.13 | 5101.1 |
| 2  | precip_1day_lp | 2 | 237 | 3.19 | 3.17 | 5049.1 |
| 3  | sm_perc | 2 | 237 | 0.86 | 0.84 | 5170.7 |
| 4  | precip_1hr | 2 | 167 | 0.99 | 0.89 | 3256.6 |
| 5  | precip_1hr_lperc | 2 | 167 | 0.87 | 0.69 | 3260.7 |
| 6  | precip_1day_lperc+sm_perc | 3 | 237 | 3.83 | 3.78 | 5018.1 |
| 7  | precip_1day_lperc:sm_perc | 2 | 237 | 3.72 | 3.70 | 5021.6 |
| 8  | precip_1day_lperc∗sm_perc | 4 | 237 | 3.94 | 3.89 | 5014.1 |
| 9  | (precip_1day_lperc:cluster)+(sm_perc:cluster) | 9 | 237 | 3.98 | 3.83 | 5022.1 |
| 10 | precip_1day_lperc∗sm_perc∗cluster | 16 | 237 | 4.25 | 3.52 | 5021.9 |
| 11 | (precip_1day_lperc∗sm_perc)+ftc | 5 | 237 | 3.97 | 3.91 | 5014.7 |
| 12 | (precip_1day_lperc∗sm_perc)+(ftc:cluster) | 12 | 237 | 4.09 | 3.91 | 5020.2 |
| 13 | precip_1day_lperc∗sm_perc∗ftc | 8 | 237 | 4.03 | 3.79 | 5017.5 |
| 14 | (precip_1day_lperc∗sm_perc)+ftc+precip_1hr | 6 | 139 | 5.59 | 5.11 | 2494.1 |
| 15 | (precip_1day_lperc∗sm_perc)+ftc | 5 | 139 | 5.59 | 5.24 | 2492.2 |
| 16 | (precip_1day_lperc∗D_perc)+ftc | 5 | 237 | 4.16 | 4.06 | 5004.8 |

lead to better results than local percentiles (models 4 and 5). The ranking between all three predictors suggested by models 1–5 is confirmed if the analysis is repeated using only the 139 events that took place during the period covered by all predictor datasets. The cross-validated logarithmic skill score for this short period is 4.38, 1.33 and 0.76 for precip_1day_lp, sm_perc and precip_1hr, respectively (not included in the table). Models 6–8 reveal that considering soil moisture in addition to the local percentile of daily precipitation improves the statistical model, with the best result obtained by using both variables individually as well as their interaction term (model 8). $LSS_{cv}$ can be further increased by adding the binary information of the occurrence of a freeze–thaw cycle in the previous 9 d to the set of predictors (model 11). Adding the binary cluster information (models 9, 10, 12) has the effect of fitting individual $b_i$ coefficients (Eq. 5) for each cluster. One would expect a better model performance if the different geological regions represented by the clusters would respond differently to the meteorological or hydrological triggers. Comparing the $LSS_{cv}$ for models 9 and 10 to that for model 8 and the $LSS_{cv}$ of model 12 to that of model 11 shows that this is not the case here. Model 10 demonstrates the importance of cross validation. This model exhibits the highest number of regression coefficients resulting in an LSS higher than for model 8. The $LSS_{cv}$ is, however, lower than for model 8, indicating that the LSS improvement is achieved by overfitting. At first sight it seems that including hourly precipitation considerably improves the statistical model as the $LSS_{cv}$ in model 14 is higher and the AIC lower than in model 11. It must be kept in mind, though, that the radar climatology is still comparatively short, and fits including hourly precipitation are based on a small subset of rockfall events, making a direct comparison of $LSS_{cv}$ and AIC with model 11 impossible. Therefore, the result of model 11 applied to the data subset used to fit model 14 is summarised in line 15 of Table 1. It shows that the increase in $LSS_{cv}$ and decrease in AIC seen for model 14 has to be attributed to the shorter time series, and the inclusion of hourly precipitation does not improve the statistical model.

An encouraging result, with respect to facilitating the analysis of climate scenario simulations, was obtained when substituting the across-site percentiles of modelled relative soil moisture used in the logistic regression model 11 with across-site percentiles of $D$ ($D$_perc, Eq. 1). In order to use $D$ as a proxy for pore water, it was accumulated over a period of time. The optimal number of days for the accumulation period was determined by successively reducing the length of the period starting at 2 weeks. The logarithmic skill score of the logistic regression model increased with decreasing duration and reached a plateau at 5 d (Fig. 5). With this accumulation period, we obtained the results shown on line 16 of Table 1. The cross-validated logarithmic skill score of that model is 4.06; thus model 16 outperforms model 11.

In addition to the combinations shown in Table 1, it was also investigated whether the regression coefficients depend on the slope angle at the event site. For this we downloaded the Copernicus digital elevation model (DEM) at 25 m horizontal resolution and calculated the slope angle at the rockfall locations using the methodology proposed by Horn (1981). The slopes at the event sites calculated using the DEM data appear plausible at many of the sites. There are, however, also locations for which we determined a slope

angle of only a few degrees, which is inconsistent with the occurrence of rockfall events. Possible explanations could be an insufficient spatial resolution of the DEM or the possibility that the slope was altered by the event and is therefore no longer captured in the DEM dataset representative of the year 2011. In addition to this, for large-scale rockfall events it is difficult to determine the exact location at which the slope needs to be estimated. Overall, including the slope angle calculated using the DEM as an additional parameter in the logistic regression model did not improve the results.

In summary, Table 1 shows that the best results are obtained from the logistic regression model 16, which is expressed in Eq. (8). The corresponding regression coefficients are $\beta_0 = -10.48$, $\beta_1 = -2.969 \times 10^{-3}$, $\beta_2 = -1.413 \times 10^{-2}$, $\beta_3 = 0.435$ and $\beta_4 = 4.053 \times 10^{-4}$.

Model 16 (Eq. 8) can now be used to predict changes in rockfall probability valid on average for specified changes of the meteorological conditions and the pore-water preconditions. The response of the rockfall probability to variations in the local daily percentile of precipitation and the percentile of the pore-water proxy $D$ is depicted in Fig. 6. The probability $\frac{es}{n}$ used as a reference to calculate the logarithmic skill score is marked with a horizontal line. As the rockfall database is not comprehensive, this value should not be interpreted in absolute terms. With $D$ and local daily precipitation set to median values (Fig. 6a), rockfall events can be expected to appear with approximately climatological probability. Less (more) precipitation leads to a probability below (above) climatological average. Increasing the local precipitation from the median to its 90th percentile approximately doubles the probability of a rockfall event. The amount of precipitation associated with the 50th percentile varies between 1.2 and 6.8 mm and between 6.5 and 31.2 mm for the 90th percentile, depending on the site. The occurrence of a freeze–thaw cycle in the previous days increases the probability of an event by about 50 %. Precipitation becomes more effective when the pore-water amount is high (Fig. 6b). When $D$ is at the 95th percentile, increasing precipitation from its median to its 90th percentile makes rockfall events almost 4 times more likely. This dependence of the slope of the probability density function for precipitation on $D$ (and for $D$ on precipitation) is made possible by the inclusion of the interaction term between precip_1day_lperc and $D$_perc in Eq. (8). The logistic regression model suggests that the influence of pore water on rockfall probability is on average less pronounced than the influence of daily precipitation. At most sites an increase in pore-water amount in the absence of strong precipitation has hardly any effect (Fig. 6c).

In terms of event numbers, the combination of $D$_perc $\leq 50$ and precip_1day_lperc $\leq 50$ includes 42 % of all days but only 25 % of all events, while the combination $D$_perc $\geq 90$ and precip_1day_lperc $\geq 90$ includes only 2 % of all days but 19 % of all events (Table 2). Combinations of high (low) precipitation percentiles with low (high) $D$ percentiles rarely occur. The climatological frequency of

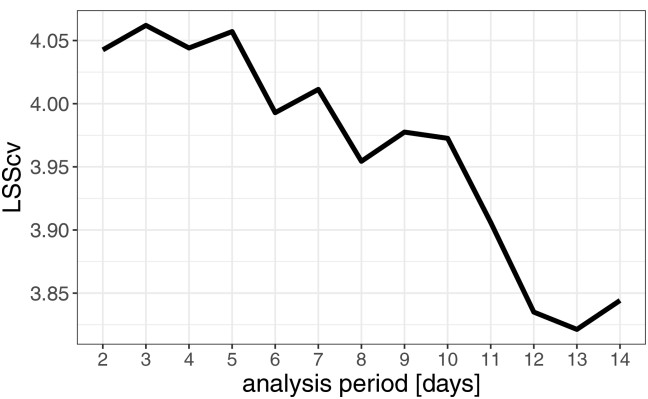

**Figure 5.** Dependence of the cross-validated logarithmic skill score on the accumulation period of $D$_perc.

rockfalls in the Elbe Sandstone cluster ES for the combination $D$_perc $\leq 50$ and precip_1day_lperc $\leq 50$ indicates that this cluster includes a higher number of events not associated with a meteorological trigger than the other clusters.

## 5 Discussion

In this study, a statistical model was developed that is able to describe changes in the probability of rockfall events in Germany that can be expected under different meteorological and hydrological conditions. It is important to keep in mind that a statistical relationship is not proof of a cause-and-effect relationship. As rockfall occurrence in Germany exhibits a seasonal cycle with a maximum in January (Rupp and Damm, 2020), it is easy to establish a statistically significant but physically incoherent relationship to any unrelated variable with a similar seasonal cycle. To account for this problem, we only included variables for which a physical relationship to rockfall events has already been established in previous studies for other sites (see introduction for details). Additionally, there is no guarantee that the sampling locations are representative for Germany as a whole. In order to investigate to what extent the model depends on the region that is investigated, we defined three study areas characterised by dense spatial clustering and high temporal data homogeneity and evaluated if the statistical model improves when the regression coefficients are allowed to differ between the clusters (models 9, 10 and 12). It was found that including the cluster information did not improve the model. This provides some reassurance that our approach to develop a single statistical model for all German low mountain ranges is reasonable. It can be assumed that the model can also be applied to neighbouring low mountain regions in central Europe with similar climatological and geological conditions.

The logarithmic skill score used to evaluate the fit of the statistical model describes the percentage improvement over a model that always predicts a climatological probability for

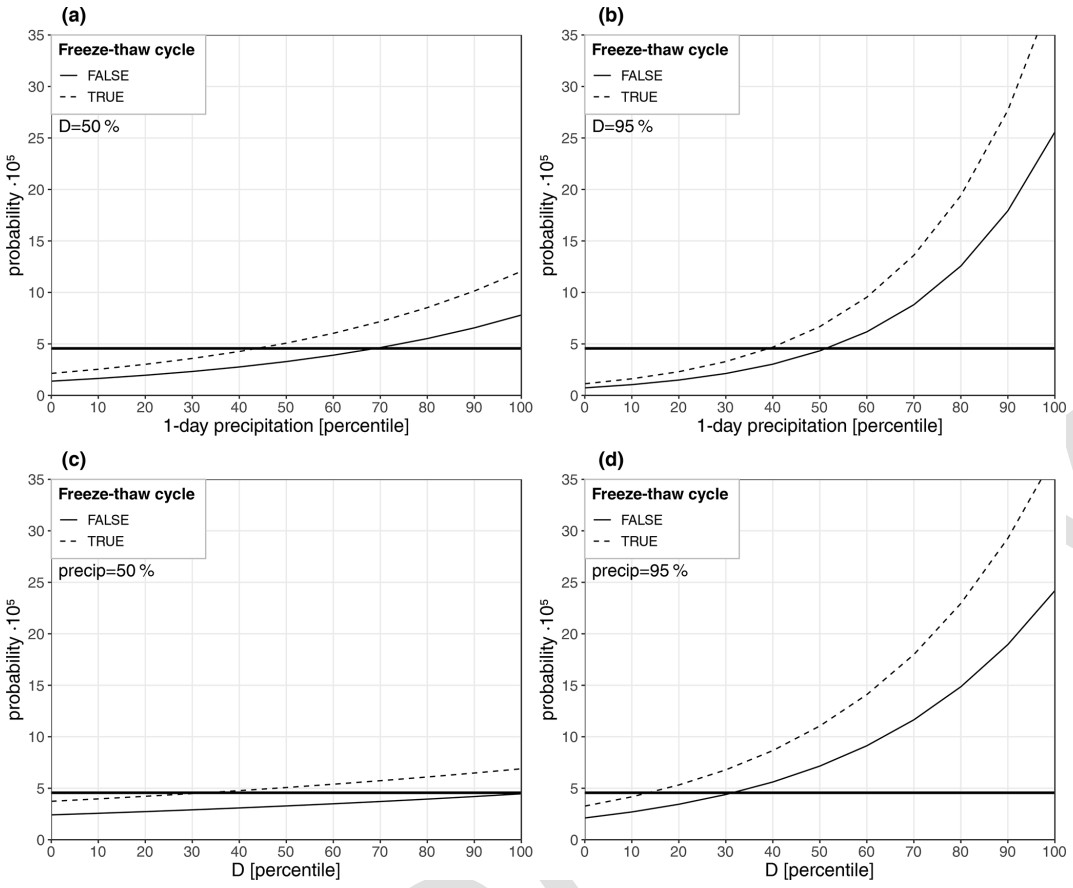

**Figure 6.** Probability for rockfall predicted by the logistic regression model. A dashed (solid) curve denotes the result for situations with (without) the occurrence of a freezing episode in the previous 3 weeks. The horizontal line marks the climatological probability. **(a)** Probability as a function of the local percentile of daily precipitation. Constant median (i.e. 50th percentile) of the pore-water proxy $D$ (i.e. precipitation minus potential evaporation for the previous 1-week period). **(b)** Probability as a function of the local percentile of daily precipitation. Constant 95th percentile of $D$. **(c)** Probability as a function of the percentile of $D$. Constant median (i.e. 50th percentile) daily precipitation. **(d)** Probability as a function of the percentile of $D$. Constant 95th percentile daily precipitation.

**Table 2.** Percentage of days with combinations of precip_1day_lperc and $D$_perc percentiles below 50 or above 90 (days). Percentage of rockfall events occurring for these percentile combinations is specified for all events (events) and separately for the events belonging to the clusters ES, HN and HR.

|  | precip_1day_lperc $\leq$ 50 | | precip_1day_lperc $\geq$ 90 | |
|---|---|---|---|---|
|  | D_perc $\leq$ 50 | D_perc $\geq$ 90 | D_perc $\leq$ 50 | D_perc $\geq$ 90 |
| % days | 42 | 4.1 | 1 | 2 |
| % events | 25 | 2.5 | 0 | 19 |
| % ES | 42 | 0 | 0 | 12 |
| % HN | 23 | 3.2 | 0 | 19 |
| % HR | 17 | 0 | 0 | 25 |

rockfall events. The skill score of our model is just over 4 % and improves to more than 5 % if only the last 20 years are used for model fitting. A value of 4 % appears to be not much, but it has to be interpreted keeping the physics of rockfall events in mind. A rockfall event can only be triggered if the slope is predisposed, after many years of weathering. Be-

cause of this, most of the time strong rainfall in an area with high soil moisture or pore-water preconditions remains without consequences (i.e. false alarms). Prediction errors (i.e. missed alarms) may also stem from events triggered by non-meteorological mechanisms or processes not captured by the chosen predictors. This seems to be the case for some events

in the Elbe Sandstone cluster ES. The model skill obtained using the selected meteorological–hydrological parameters as predictors, however, suggests that non-meteorological influence and missing predictors seem to be subordinate factors in the rockfall process for the selected study regions.

As this model was developed for the purpose of detecting changes in rockfall probability in climate scenario simulations, the low skill score on account of the overall low probability for rockfall does not pose any problems. For a warning system, the number of false alarms would be too high. This limitation could only be overcome by including information on the predisposition in the statistical model. Unfortunately, this is not feasible as it would be far too expensive to monitor every slope operationally. Nevertheless, the concept of using a logistic regression model instead of fixed thresholds would also have advantages for warning systems. The probability for rockfall relative to a baseline climatology could be determined with Eq. (8) from the output of an operational weather forecasting model. Which values should be regarded as a low, medium or high risk could be defined by the operator using the model. With a predefined matrix constructed from combining meteorological observations with event numbers (such as in Table 2), this flexibility does not exist.

We found that daily precipitation is the most important factor to trigger rockfall events in Germany. The best fit for the statistical model was obtained when using local percentiles rather than across-site percentiles (not shown) or absolute values. A possible interpretation could be that most rock slopes are balanced under normal climate conditions but can become unstable in the presence of above-normal precipitation amounts. The presence of freeze–thaw cycles increases the probability by approximately 50 %. Pore water on its own is unlikely to trigger a rockfall event. It can weaken porous material, making it more susceptible to a trigger like precipitation. The fact that both simulated soil moisture and $D$ improved the statistical model confirms that these variables can be used as a first-order substitute for all relevant types of subsurface moisture, such as cleft water and water in rock pores.

Quantitatively, our findings are in contrast to those of D'Amato et al. (2016) and Bajni et al. (2021), who reported the most important rockfall trigger to be the freeze–thaw cycle for middle mountain ranges in France and for the Italian Alps, respectively. Macciotta et al. (2017) named precipitation and freeze–thaw cycles as the most likely dominant factors but refrained from ranking their importance. The differences between the studies are site-specific and stress the fact that meteorological parameters and thresholds are spatially heterogeneous and need to be determined for each region individually. Unlike Sass and Oberlechner (2012), we were able to establish a robust relationship between precipitation, a temperature-related predictor (freeze–thaw cycles) and rockfall, suggesting that daily observations with a spatial resolution of a few kilometres are sufficiently accurate to capture the micro-climatic conditions.

## 6  Conclusions

Using a rockfall dataset for Germany, it was possible to build a statistical model that is able to quantify changes in rockfall probability in response to changes in pore water and meteorological factors identified in geophysical studies as potential triggers for rockfall events. The model can be regarded as representative for the low mountain ranges in Germany. It can also be used in other central European low mountain regions with similar climatological, hydrological, geological and topographical characteristics for which no customised modelling approach exists.

The model was developed in order to be applied to climate change simulations, with the aim of determining if the probability of rockfall events can be expected to change in response to global warming. Applying the statistical model to climate simulation output is facilitated by the fact that the model works with percentiles for most predictors. Thus, only temperature for the evaluation of freeze–thaw cycles needs to be bias corrected. In addition, the complex simulation of soil moisture can be substituted by a pore-water proxy (i.e. accumulated precipitation minus potential evaporation), which can be easily calculated from climate model output.

For application in climate change studies, it is important that the statistical model considers the interaction between the triggering factors as these are expected to show opposing trends. While heavy precipitation is likely to increase in the future (IPCC, 2014), a decrease in the number of frost days dependent on altitude can be expected with the projected increasing global temperatures (IPCC, 2014). Climate projections for aridity in central Europe depend on location and season (Samaniego et al., 2018). Thus, studies considering only single factors might over- or underestimate the response of rockfall to climate change as the interaction of the factors can amplify or diminish the signal.

*Data availability.* The meteorological data used in this study are freely available. After registration the E-OBS dataset can be downloaded from https://www.ecad.eu/download/ensembles/ensembles.php (European Climate Assessment and Dataset, 2022). REGNIE is available from https://opendata.dwd.de/climate_environment/CDC/grids_germany/daily/regnie/ (DWD Climate Data Center, 2021) and RADKLIM from https://doi.org/10.5676/DWD/RADKLIM_RW_V2017.002. Information on the rockfall events can be found in the supporting material of Rupp and Damm (2020). The Copernicus digital elevation model is freely available from https://land.copernicus.eu/imagery-in-situ/eu-dem/eu-dem-v1.1 (Copernicus Land Monitoring Service, 2016).

*Author contributions.* KMN conducted the statistical analyses and prepared the draft manuscript. SR and BD collected and analysed the rockfall data. SR prepared Fig. 1 and provided the rockfall data description. TMK wrote the introduction and BG conducted the soil

moisture simulations. BD and UU supervised the project and provided advice and feedback in the process.

*Competing interests.* At least one of the (co-)authors is a member of the editorial board of *Natural Hazards and Earth System Sciences*. The peer-review process was guided by an independent editor, and the authors also have no other competing interests to declare.

*Acknowledgements.* This study was funded by the Federal Ministry of Education and Research in Germany (BMBF) through the research programme ClimXtreme (FKZ (grant nos. 01LP1903A, 01LP1903K and 01LP1903E)). The work used resources of the Deutsches Klimarechenzentrum (DKRZ) granted by its Scientific Steering Committee (WLA) under project IDs b1152 and bm1159. We acknowledge the E-OBS dataset from the EU-FP6 project UERRA (http://www.uerra.eu, last access: 13 August 2021) and the data providers in the ECA&D project (https://www.ecad.eu, last access: last access: 13 August 2021). We would also like to thank the two anonymous reviewers whose constructive comments helped to improve the paper.

*Financial support.* This research has been supported by the Bundesministerium für Bildung und Forschung (grant nos. 01LP1903A, 01LP1903K and 01LP1903E). TS5

We acknowledge support from the Open Access Publication Initiative of Freie Universität Berlin.

*Review statement.* This paper was edited by Paola Reichenbach and reviewed by two anonymous referees.

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

**Remarks from the language copy-editor**

CE1    The Oxford comma must be used for clarity in the case of a complex list (i.e., when an item includes "and" or "or"). This is standard in English and has already been made consistent throughout.

**Remarks from the typesetter**

TS1    We have changed the asterisk to a cdot in the previous equation.

TS2    We have changed the asterisk with a cdot.

TS3    We have used another command and noticed that some cdots were not adjusted during initial typesetting.

TS4    Thank you for your feedback. Meaning and content changes, including changes to equations, should be reviewed by the editor before being implemented in the proofreading stage. Please reassess if these changes are strictly necessary before taking this step. For more information, please see our proofreading guidelines at: http://publications.copernicus.org/for_authors/proofreading_guidelines.html. If you want us to change the equations, please prepare an explanatory document (doc or pdf) which we can send to the editor via our system.

TS5    I was referring to the normal statement not the institutional statement. If you want to update the acknowledgements or this section, please let us know.

TS6    The rule is to cite the reference the way they have been published. If the title was published in capitals, it is fine to leave it. I hope this answers your question.

TS7    I have updated the year.

TS8    We have moved it to the correct alphabetical order. Sorry for this oversight. It is up to you if you want to rename it. Let me know what you decide.