# Peer review of "Quantification of meteorological conditions for rockfall triggers in Germany"

_Natural Hazards and Earth System Sciences, 2021_

## Author Response (AR1)

**Description of revisions**

This document lists all reviewer comments and describes the revisions related to them.

**Review 1:**

The manuscript of Nissen et al. deals with the set up of a logistic regression model to derive the probability of occurrence of rockfalls in Germany, based on meteorological and hydrological variables. The paper is interesting and the writing is fluent and clear. I enjoyed reading it. Regarding results, the authors are able to quantify the impact of increasing rainfall and increasing subsurface water (i.e., pore water, water in fractures) in terms of variations of probability of occurrence of rockfalls. Despite its general good quality and interesting findings, I believe there are certain aspects that need improvement. While presenting the results of both the selection of predictors procedure and the logistic regression model, often some aspects are not shown. Since the manuscript is very concise and there isn't an excessive number of figures, I suggest to show some additional detail.
We added 2 figures to the main manuscript. In addition, we decided to prepare a supplement which includes 2 more figures.

Results are clearly presented, expressed in quantitative terms and well-linked to the performed analysis. However, I believe that presenting them with an additional operative perspective could give a lot of added value to the manuscript. In which conditions did most of the rockfalls occurred? Above which percentile of precipitation, subsurface water content? We added a table which shows the numbers for a selection of 4 combinations.

Would it be possible to translate the probabilities of occurrence in a matrix of no-, low-, mid- and high-hazard? See also specific comments #23, #24, #27.
We added a statement to the discussion explaining the benefits of using a statistical model instead of working with a threshold matrix.

In terms of presentation, to favor readability, I suggest to consider a more rigorous and classic structure with Intro – Study Area and Data – Methods – Results – Discussions – Conclusions. Now, each step is presented with methods and results together. I got lost just a couple of times, not compromising the general understanding of the work, but I would find it easier to follow with the suggested modification.
We restructured the manuscript.

In addition, dealing the paper with rockfalls, I suggest a thorough check of the use of terms dealing with subsurface water (e.g., pore water, water in fractures, soil moisture, subsurface moisture etc.), since it is not always clear to what exactly the authors are referring to.
We homogenised the usage of the terms.

Discussions need to be extended with at least an additional paragraph comparing the results with previous literature on the topic (see comment #25)
We added a paragraph to the discussion.

1. L9-10: Precipitation minus potential evaporation corresponds to what was defined as simulated soil moisture or parameterised pore water? Also, it was stated that moisture observations are not available so which was the assumption made to evaluate the performance of the proxy?
   We added that we evaluated its performance as a predictor for rockfall.

2.  L18-20: I believe that the described conditions can be most common between the end of the winter season and the beginning of Spring (late March-April) but it could be an Alpine bias of mine.
    This is correct and we added "and beginning of spring"

3.  L27-28: Consequently [...] site specific. The sentence probably needs a couple of example references.
    We added two references concerning this statement.

4.  L34-35: in this sentence, is promote used to say that the weathering mechanisms are preparatory-predisposing factors?
    Yes, exactly.

5.  L36: wetting and drying of porous rocks (esp. argillaceous) à I am pointing out this sentence as an example but I think it should be clarified throughout the manuscript how the term porosity (pore) is used (other example in the abstract, L4). Is it matrix porosity only or does it include discontinuities (joints, fractures, etc)? I suggest to specify it in the text. In this specific case, if talking of matrix porosity, maybe it is better to use sandstones as example rather than argillaceous rocks.
    In this specific case, it means matrix porosity, we clarified the sentence and referred to sand- and siltstones specifically (the same as the associated citation). Further usage of pore water does include discontinuities, we specified this for clarification at the appropriate location.

6.  L40-44: earthquakes could be added to the list, although probably not relevant in Germany.
    Thank you, we added earthquakes, which are not completely irrelevant even in Germany.

7.  L52: water in rock cracks can act as a weathering agent through both physical and chemical processes. I would expect not only water presence but also wet-dry cycles causing repeated high water (over-)pressures in fractures to reduce the rock mass strength (i.e., weather the rock mass). I suggest to add some additional explanation to the sentence.
    Indeed, we formulated it subjunctively, as it is an example of a possible negative feedback (not a certain one) on rockfalls from changing metrological conditions.

8.  L58-72: Rockfalls have the year of occurrence but the model is based on hourly, daily and weekly data. How was it possible to fit the model? How could you relate specific values of the meteorological and hydrological variables to the occurrence of the different rockfalls?
    For 343 events the day is known. We specified that in the text.

9.  L94-96: I would move this last paragraph towards the end of section 2.1 and introduce Fig. 2 there.
    We moved the paragraph.

10. L112-119: Please add some details regarding the model. In particular, it is mentioned that it is calibrated using gauge measurements; are these gauge measurements soil moisture sensors or else? Which is the time resolution of the model? Also, the model allows simulations through the entire column from the surface to a depth of approximately 1.8 m; within this depth, does it allow to distinguish between actual soil and rock? Is the water infiltration process modelled in the same way for both materials?
    The hydrological model mHM was calibrated based on discharge time series. Within mHM, different soil types with different soil layers based on a soil map that is used as

input are distinguished. Thus, each soil type varies in its characteristics such as hydr. conductivity. The infiltration from the surface into the soil depends on the soil characteristics. We extended the description in the manuscript.

11. L120: what is meant for operationally available? If I think of a model used for operational purposes I would refer to numerical weather predictions (short or midrange, i.e. few days) rather than climate models (decades).
The term operationally was misleading. The model is supposed to be applied to climate simulations. The sentence was rephrased.

12. L133-134: Please specify the accumulation periods that were tested. How was the performance of the different accumulation periods evaluated? The fact that the weekly period behaved best is a result.
The fact that we tested periods between 14 and 6 days was added. A figure was prepared for the supplement.

As stated in the general comment, I suggest to consider the possibility of re-organizing the manuscript with a more rigid and classic structure (Intro- Study area – Methods – Results – Discussion – Conclusion).
We restructured the manuscript and moved the result about the best accumulation period into the results section.

13. L135-138: it is true that with this approach trivial areas (e.g., flat terrain, no rock) were excluded but it could be that potential unstable areas were excluded too. It is a reasonable approach to set-up the model but if predictions are necessary the areas should be filtered based on other terrain and land-use data. I suggest to motivate it explicitly, including the part on the exclusion of the grid boxes with events occurred in periods not covered by meteorological/hydrological data.
The statement was rephrased: A relationship between the trigger and the event can only be established for sites and periods for which both elements are known.

14. L145: How are the range of values selected?
The bins have to contain an equal number of observations. This was added to the text.

15. L149-164: Mostly results, therefore same suggestion as comment #12. Also, the manuscript does not have an excessive amount of Figures, so why not showing part of the results?
Results were moved to results section and additional figures prepared for the supplement.

16. L179-180: is resolution intended as spatial, temporal or both? In any case I miss the direct link with model comparison. I would expect two models to be comparable if the input (training) data correspond, while in this section it is said that they might change according to the data used (daily precipitation, soil moisture, hourly precipitation).
The text related to horizontal resolution, event sites and the comparability of statistical models was rephrased and a link with the model comparison added.

17. L209: it is not clear to me how the cluster predictor works. Can you please explain with additional detail the rationale of including it in the logistic regression model? From L199-201 I understood that the cluster number was just used to subsample the available data and split them in training and test sets.
When using the cluster predictor a different set of fitting coefficients ($b_i$) is determined for each cluster. A better model performance would be expected if the different

geological regions represented by the clusters would respond differently to the meteorological/ hydrological triggers. We rephrased the passage.

18. L212-213: Why considering all the interaction terms? Physically, what do you expect the product of two terms can explain that their addition does not?
We added an explanation on this issue to the section describing Fig. 6 (previous Fig. 4).

19. L218-219: The introduction of AIC belongs to methods.
The AIC introduction was moved.

20. L225: Similarly to comment L212-213, what could be the physical meaning of the interaction term between the local percentile of daily precipitation and soil moisture? Why was it decided to include it in the model?
We added an explanation on this issue to the section describing Fig. 6 (previous Fig. 4).

21. L227-228: Where is it possible to see that the customisation of the model for regions does not improve the performance?
We extended the explanation in order to clarify which models need to be compared.

22. L242: Where is it possible to see it?
We added another figure (new Fig. 5) to show this. For the preparation of the figure we extended the analysis to periods shorter than 6 days and found a slight improvement for 5 days. We replaced numbers and figures determined using the weekly period with numbers and figures for the 5-day period. The effect on the results is negligible.

23. L258-259: Among the 237 events used to fit Model 16, given D at its median value or below, how many events occurred for conditions of precipitation below the median and how many for conditions below the 90th percentile? Is it the same in all the three study areas?
We added a table which shows the numbers for a selection of 4 combinations.

24. L262-263: Same as above but given D at its 95th percentile or below.
We added a table which shows the numbers for a selection of 4 combinations.

25. L267-298: In the discussion section, at least a paragraph should be dedicated to the comparison of results with analyses of previous studies. Perhaps the subject is not identical, but in the introduction several studies are cited that discuss causes and relationships of climate and hydrological variables with rockfalls (e.g., Bajni et al., 2021; D'Amato et al., 2016; Macciotta et al., 2017; Saas and Oberlechner, 2012). I think it could be interesting to know how your results, or the general indications given by your results, compare to those of these studies and maybe other similar ones.
We added a paragraph to the discussion.

26. L275: Data regarding other regions of Central Europe are not presented. I would phrase it a bit more carefully saying that given the similar climatological, hydrological, geological and topographical characteristics the model could be applied in other low mountain areas of Central Europe with success. An evaluation of its performance would still be needed. The same comment applies to the conclusions (L303) and the title.
We changed the title and rephrased the two passages.

27. L277-289: in this paragraph, false alarms and prediction errors are discussed. However, in the results there isn't a real attempt to set probability thresholds and quantify these values. I know how difficult it is but based on the derived models and the recorded

occurrences, could you suggest combinations of values to define no (low) hazard, medium hazard, high hazard? This comment is linked to comments #23 and #24 too. *We rewrote the paragraph and added a statement on the benefits of using a statistical model instead of fixed thresholds.*

28. L312: frost days will decrease, but freeze-thaw cycles could increase at specific elevations. I'll suggest to phrase it more carefully.
*The sentence was rephrased.*

**Minor editorial comments**

L5: both for the day of the event and the days leading up to it. *Corrected.*
L36: dissolution in carbonatic rocks *Corrected.*
L49: all climatic factors that promote *Corrected.*
L53: a statistical model that *Corrected.*
L58: rockfall data that. Please check throughout the manuscript the use of which/that. *Was done.*
L61: database, which [...]. The database mainly covers the last 200 years *Corrected.*
L62: Information on 670 rockfall events are included in the landslide dataset. *Changed.*
L68: while the remaining *Corrected.*
With the majority of them (n=621) recorded from *Corrected.*
L105: 1 km x 1 km or 1 x 1 $km_2$ *Changed.*
L258: Less/more precipitation leads to a below/above *Corrected.*

**Review 2:**

**General comments**

The manuscript proposes a statistically-based approach to quantitatively assess the impact of a set of meteorological and hydrological variables on rockfall occurrence in some selected low-mountain regions in Germany. These variables are thus considered as potential triggering factors and have been analysed for the event day and for the days before. The authors conclude that the logistic regression-based model used is able to detect changes in the probability of rockfall occurrence in the study area. Precipitation at daily scale turn to be the main triggering factor and a 5-parameters model based on the interaction of daily precipitation, freezing-thaw cycles and increase in sub-surface water is the most appropriated in terms of skill.

The manuscript represents a valuable and innovative contribution to the understanding of climate variables-related impact on the hydrogeological risk. Outcomes of the research are very interesting and the paper is in general well-written and fluent. I'm not a mothertongue but I think that the English is good. Nevertheless, I personally think that some major changes are needed and would improve the overall quality of the paper.

First, I think that he paper should be restructured in a more standardized and classic way to improve readability, in particular the data and methods sections. I suggest to state clearly each part e.g., Data, Methodology, Results etc. The manuscript as is mixes together data, methods and results and I must admit that I encountered some difficulties in following the text.
*We restructured the manuscript.*

The description of data and methods (e.g., methods to simulate soil moisture and pore water proxy) is not sufficiently complete. Further specification and clarification should be included throughout the text. Adding some figures related to not shown analysis could be of help in this (see Specific comments).
*We included 2 additional figures and compiled a supplement containing another 2 figures. They are listed under the specific comments.*

The Discussion as is lacks of comparisons with similar studies on the same topic (i.e., relations among climate, hydrological variables and rockfall occurrence) to assess how they differ or agree.
We added a paragraph to the discussion.

Other specific comments are listed hereinafter. Moreover, I think it is a bit ambitious to say that the model is valid for the entire Central Europe as stated in the title as well.
We rephrased the text and changed the title.

**Title**
I suggest to revise the title. The considered dataset is not representative of Central Europe actually.
The title was changed

**Abstract**
L 10: Maybe it should be clarified that precipitation minus evapotranspiration is a pore water proxy (as an alternative to simulated soil moisture).
We rephrased the passage.

**Introduction**
L 33: Is the term "promote" related to predisposing condition for rockfall occurrence? Check the meaning throughout the text.
Yes, exactly.

L 45-55: Actually, studies focusing on statistically-based approach to assess the linkage between climate forcing and rockfall/landslide occurrence are increasing worldwide and especially in mountain-areas like e.g., the one investigated in this work. I suggest to discuss further this point and mention some relevant works in this context, highlighting how this study adds value and contributes to shade light on climate and hydrological related trigger mechanisms.
We made an addition to the introduction and, accordingly, the discussion.

**Meteorological and hydrological variables**
L 112- 119: I would add some further details on the methodology adopted. What's the resolution of the model? Can the model distinguish between different types of lithologies across the entire column?
The model simulates with a daily resolution and calculates all components of the water cycle for each day.  Horizontally, the model has a fixed grid size of 5 km. Vertically, the model includes 6 levels from the surface up to the here considered depth of 1.8m. The hydrological model considers different soil types based on a soil map that is using as input for the modelling. Each soil type consists of different layers and has thus different soil characteristics.
We extended the description in the manuscript.

L 114: In general, the use of terms related to soil moisture ad sub-surface water throughout the text looks a bit misleading to me, since it is not always clear to what the authors refer to.
We checked the manuscript and homogenized the terms.

Figure 1: Please add the meaning of the cluster in the caption.
Was added

L 120: It is not clear if the authors refer exactly to climate scenarios or to a shorter-term prediction. Before, there was no reference on the use of climate change scenarios for this analysis.

The term operationally was misleading. The model is supposed to be applied to climate simulations. The sentence was rephrased.

L 120: Not clear how the authors validated the pore water proxy if soil moisture information are available only for some sites and simulations. In general, I think that the authors should provide more details on the methodology and procedures used to calculate the soil moisture and pore water proxy.
We don't validate simulated soil moisture and the pore water proxy. We only compare their performance as predictors in the statistical model. The description in the abstract might have been misleading and was rephrased.

L 128-129: Please rephrase.
The sentence was rephrased

L13 L 145: How the authors select the range of values for each variable?
The bins have to contain an equal number of observations. This was added to the text.

L 149: These are part of the results.
We restructured the paper

L 155-164: This is a mix of results and methods. Please see the general comment.
We restructured the paper

L 156: Considering time spans before the rockfall events is useful not only for thawing process, but for the antecedent moisture condition as well. It looks like that the time span including the days leading-up to the failure have been considered only for thawing process while, as can be derived from L 134, different accumulation periods prior to the failure have been investigated also for pore water. Please clarify better this point.
We added another figure (new Fig. 5) which shows the time lags analysed for the calculation of the pore water proxy. This should clarify the point. For the preparation of the figure we extended the analysis to periods shorter than 6 days and found a slight improvement for 5 days. We replaced numbers and figures determined using the weekly period with numbers and figures for the 5-day period. The effect on the results is negligible.

L 157: What time spans did the authors consider?
Time spans analysed for freeze-thaw cycles can be seen in the new Figure 4.

L 161: It is not very clear to me how the authors could compare the importance of such variables if the number of rockfalls and grid change depending on the spatiotemporal resolution of the respective datasets. Maybe it is worth to include the results of the consistency test here or as Supplementary Material.
An additional figure (S1) was added to the supplement.

L 165: This part seem to be more appropriate for a discussion.
The paragraph was moved to the discussion

L169: Please add some reference.
We refer to the studies cited in the introduction and added this information to the statement.

L 178: It is not very clear to me how the WOE considers the fact that more than one landslide occurs in a grid box.

We only use unique observational time series (instead of using the same time series twice if two events occurred close together). We rephrased the explanation.

L 179: I guess that the resolution is intended as spatial?
The term "spatial" was added

L 213: This point has to be clarified. Please discuss further why using all possible combinations of predictors instead of simply the sum or product could add value to the analysis.
We added an explanation on this issue to the section describing Fig. 6 (previous Fig. 4)

**Results**
L 218: AIC has not been introduced before. Please add further details in the methodology section.
AIC is explained in the new methodology section

L 238 The authors should introduce the term "across-site percentile" and clarify it.
We added the explanation to the data section.

L 246 This can be due also to the DEM resolution that could not be sufficiently higher to detect the exact location of a rockfall.
Was added to the text

L 254 AIC considers both how well the model reproduces the data and the number of variables maximum likelihood estimates of the model) used to build the model. The lower the AIC value, the better the model fits as the authors rightly state at L 220. If I rightly understood, AIC is much lower in model 15 compared to the selected one (16). Is the choice due to the number of rockfall events involved? The authors partially explained the final selection; thus, I would suggest to discuss further this point.
We extended the explanation regarding the comparability of the statistical models to clarify this point.

**Discussion**
L 270: The authors rightly state that there is no guarantee that the three locations are representative for the entire Germany but at L 275 they say that the approach can be reasonably extended in Central Europe regardless of the geographical and local settings. I think that additional analyses are needed to state this actually.
We rephrased the statement.

L 284 I would say that also the inclusion of more climate and hydrological variables could be of help in decreasing the number of missed alarms.
We rephrased the statement.

L 291 As I understood correctly, time series of different lengths have been used depending on the considered variable. To determine the percentile of the variable in the lead-up of the event with relation to the previous period, the authors use both a local and across-site percentile. Across-site percentiles could be misleading in this context. Does "across-percentile" mean that the same variable is compared across records of different weather stations with different lengths while the local percentile is referred to the local time-series?
As we use gridded data sets the length of the time series for a selected variable is the same for each grid point. The length of the time series for D and daily precipitation is the same. No change was made in the text.

L 303 Be careful about saying that the model is representative for low-mountain regions in Central Europe. I would say in Germany, at most.
We rephrased the text and changed the title.

**Technical corrections**
L 6: and instead of "as well as" Was changed
L 49: "that" instead of "which" Was corrected
L 105: Be consistent throughout the text: 1 km x 1 km 1 km2 (L 102). Was changed
L 258: "a below" instead of "an below" Was corrected